# Adolescent fertility and its determinants in Kenya: Evidence from Kenya demographic and health survey 2014

**Naomi Monari**⦿ *, **James Orwa, Alfred Agwanda**

Population Studies and Research Institute, University of Nairobi, Nairobi, Kenya

* naomi.monari@gmail.com

**Data Availability Statement:** The data underlying this study are available at https://dhsprogram.com/data under file name "KEIR72SV.ZIP". Interested researchers can replicate our study findings in its entirety by directly obtaining the data from the third

## Abstract

### Background

Adolescent fertility in Kenya is vital in the development and execution of reproductive health policies and programs. One of the specific objectives of the Kenyan Adolescent Sexual Reproductive Health (ASRH) policy developed in 2015 is to decrease early and unintended pregnancies in an attempt to reduce adolescent fertility. We aimed to establish determinants of adolescent fertility in Kenya.

### Methods

The Kenya Demographic and Health Survey (KDHS) 2014 data set was utilized. Adolescent's number of children ever born was the dependent variable. The Chi-square test was utilized to determine the relationship between dependent and independent variables. A Proportional-odds model was performed to establish determinants of adolescent fertility at a 5% significance level.

### Results

Over 40% of the adolescent girls who had sex below 17 years had given birth i.e, current age 15–17 years (40.9%) and <15 years (44.9%) had given birth. In addition, 70.7% of the married adolescents had given birth compared to 8.1% of the unmarried adolescents. Moreover, 65.1% of the adolescents who were using contraceptives had given birth compared to only 9% of the adolescents who were not using a contraceptive. Approximately 29.4% of the adolescents who had no education had given birth compared to 9.1% who had attained secondary education. Age at first sex (18–19 years: OR: 0.221, 95% CI: 0.124–0.392; 15–17 years: OR: 0.530, 95% CI: 0.379–0.742), current age (18–19 years: OR: 4.727, 95% CI: 3.318–6.733), current marital status (Not married: OR:0.212, 95% CI: 0.150–4.780), and current contraceptive use (Using: OR 3.138, 95% CI: 2.257–4.362) were associated with adolescent fertility.

### Conclusion

The study established that age at first sex, current age, marital status, and contraceptive use are the main determinants of adolescent childbearing. The stated determinants should

party and following the protocol in our Methods section. We confirm that the authors did not have any special access privileges that others would not have.

**Funding:** The author(s) received no specific funding for this work.

**Competing interests:** The authors have declared that no competing interests exist.

**Abbreviations:** KDHS, Kenya Demographic and Health Survey; TFR, total fertility rate; TF, Total Fecundity; ASFR, Age Specific Fertility Rate; AF, Age Fecundity; OR, Odds Ratio; CI, Confidence Interval; RC, Reference Category.

be targeted by the government to control the adolescent birth rate in Kenya. Consequently, delaying the age at first sex, discouraging adolescent marriage, and increasing secondary school enrollment among adolescent girls are recommended strategies to control adolescent fertility in Kenya.

## Introduction

According to World Health Organization (WHO), approximately 16 million adolescent girls give birth every year, an average global birth rate of 49 per 1000 births. Nearly 95 percent of these births occur in low and middle-income countries with a majority of the births occurring among adolescents who are less educated, poor, and rural residence [1]. Despite the universal decline in the adolescent birth rate since 1990, adolescent fertility remains high in many developing countries [2] and remains a great concern to policymakers. The highest percentage of adolescent births which is 46 percent is experienced in Sub-Sahara Africa, followed by Southern and Central Asia at 18 percent then Latin America and the Caribbean at 14 percent [3]. Sub-Saharan Africa's (SSA) adolescent birth rate is estimated at 101 births per 1,000 women [3]. African countries with the highest teenage pregnancies are Niger, Mali, Angola, Mozambique, Guinea, Chad, and Cote d'Ivoire [4].

In Kenya, the adolescent birth rate has been declining more slowly from 168 live births in 1977/78 to 96 live births in 2014 with the rate remaining high [2, 5]. In 40 years, this is a gradual decline considering the Kenyan total fertility rate (TFR) experienced a major decline between 1977–78 and 2014 from 8.1 to 3.9 births per woman and the fact that the average adolescent birth rate in developed countries stands at 19 live births per 1,000 women, a difference of 77 live births. The adolescent fertility rate in Kenya increases swiftly as adolescents advance in an age such that 3 percent of adolescents had given birth by their 15th birthday while 40 percent of the adolescent had given birth by their 19th birthday [5]. It is reported that in Kenya about 11% of the female adolescents had given birth before 20 years of age [3].

Various studies have been undertaken to establish key factors associated with adolescent fertility. Key determinants identified in these studies are current age, type of place of residence, education level, contraceptive usage, and wealth index [6–10]. However, other factors that are associated with adolescent fertility but vary from country to country and even within the countries disparities still exist were, employment status, marital status, postpartum infecundability, parents' income, religion, media exposure, the status of living with a partner and practice of sexual relations [6–10]. According to Bongaarts [11], 96% of the difference in fertility levels among societies is explicated by four proximate determinants which are the prevalence of contraceptive use, incidences of induced abortion, fertility inhibiting effect of breastfeeding, and the proportion of females married. These four variables provided a parsimonious framework with measurable and quantifiable variables.

In Kenya, a myriad of factors are associated with the escalating adolescent birth rate with limited studies identifying key factors influencing adolescent childbearing. Westoff [12] established that the rise in adolescent fertility was associated with the stall in contraceptive use among Kenyan youths [12]. The proportion of adolescents aged 15–19 years who did not approve the use of family planning increased from 13.4 percent to 22.4 percent (a 66% increase) between 1998 and 2003. In addition, the study established that fertility preferences of adolescents were changing in favor of larger family sizes and their negative attitude towards family planning was also increasing. According to [13], adolescent childbearing in Kenya was

higher in rural than in urban areas, among 18 to 19-year-olds, those with primary or no schooling, those who did not attend school, ever married individuals, and those from households in the bottom two wealth quintiles. The study also established that pregnancy was unintended in a quarter of the adolescent girls who had begun childbearing. In recent years, premarital childbearing in Kenya has been of great concern among policymakers, an aspect that is amplified by the terse observation that the majority of the young pregnant women are poor, not in a union and that even if they were, the union could be illegal.

This study had two aims, first, to find out proximate determinants of adolescent fertility outcome (aggregate analysis and disaggregation by place of residence) and second to establish select demographic, economic, and cultural factors significantly associated with adolescent fertility outcomes at an individual level within the Bongaarts framework and hence contribute to the body of knowledge. Exploring the factors associated with teenage childbearing ensures targeted programs to this vulnerable population sub-group for policymakers and to aid in reducing the teen pregnancy incidence.

## Methods

The study used the 2014 KDHS dataset which is the sixth national Demographic and Health Survey since 1989 when the first nationally representative survey of this nature was conducted. The 2014 KDHS data was collected from May 2014 to October 2014 where a sample representing all women between 15 to 49 years was interviewed. The survey collected data on women's complete birth history which makes it appropriate for this study. A sum of 31,079 women respondents participated in the survey. For this study, we included adolescent women aged 15–19 who were 5,820 (18.7%) of the total women interviewed. The operation framework used for the variables included in the study is presented in Fig 1.

### Background factors

Fig 1.

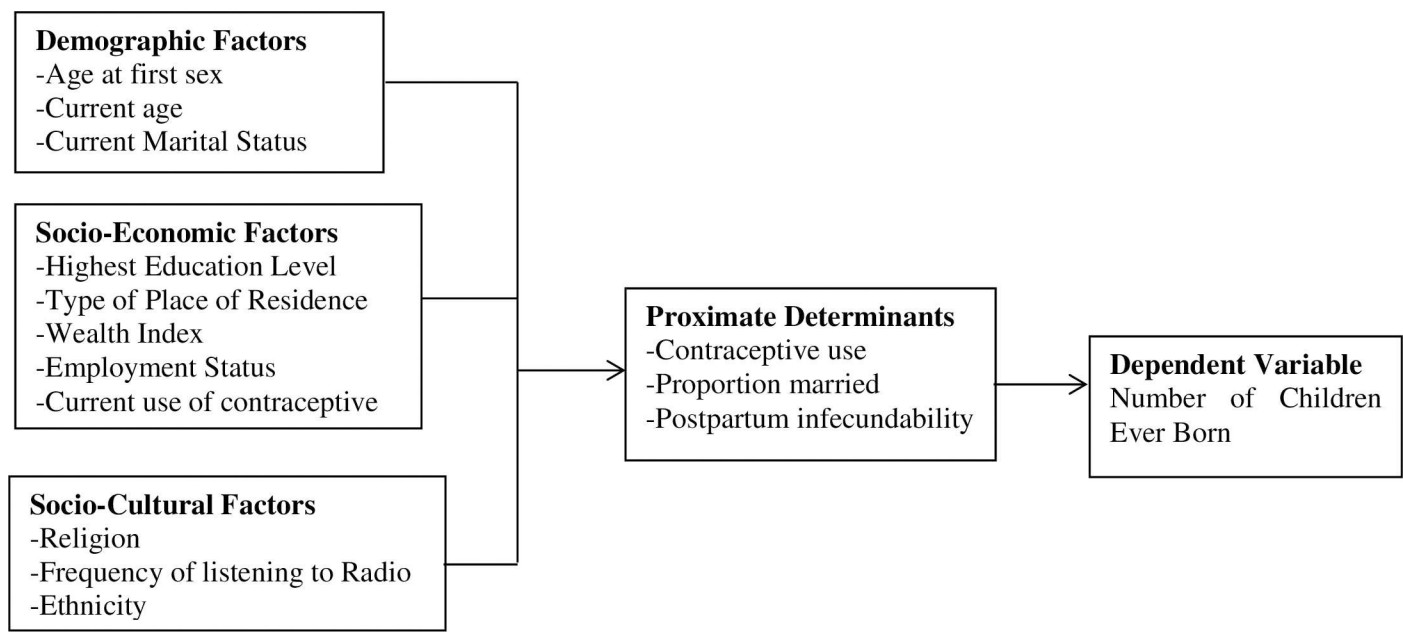

**Fig 1. Operational framework.**

## Study variables

Adolescent Fertility, the key-dependent variable was used interchangeably in the study to mean adolescent childbearing, teenage childbearing, or teenage fertility. This was the total number of children a female between age 15 and 19 had given birth to at the time of the survey. The variable was assessed by recoding the KDHS variable on Total Number of Children Ever Born among the female adolescent, 15–19 years, as (0, 1, and 2+). Independent variables included age at first sex which was the age at which a female adolescent (15–19 years) experienced her first sexual intercourse; current age, current marital status, highest education level which was the highest level of education the female adolescent had attained during the period of the survey (Higher, primary, secondary, no education), type of place of residence, wealth index, which was measured in the KDHS by creating an index from household assets including radio, TV, bicycle, car, electricity, motorbike and dwelling features such as sources of water and sanitation facilities and type of material used for roofing and construction. In the study female adolescents were grouped according to their wealth status (rich, middle, and poor). Other variables were employment status, current contraceptive use, religion, frequency of listening to Radio and ethnicity.

## Statistical analysis

The dependent variable was categorized into no child, one child and two or more children which was ordinal. We explored the variables descriptively using frequency and percentages and then compared them using the Chi-square test to find out specific significant demographic, economic, and cultural factors associated with the dependent variable. The ordinal regression model was used to model the effect of explanatory variables on the predictor variable. The proportional odds model was preferred due to the ordinal nature of the outcome variable of interest. In addition, the Bongaarts model on proximate determinants of fertility was also utilized to identify the factors.

**Proportional-odds cumulative logit model.** The model overcomes limitations of the chi-square test as it shows the magnitude of the relationship between the response and the independent variables in terms of the odds ratio. There were three groupings of the dependent variable, which was similar to two binary responses i.e. (i) 2+ children versus 1 child or no child and (ii) 2+ children or 1 child versus no child. There was thus a cut-off (threshold) at 2+ children the first logit and another at 1 child, forming the second logit. The model simultaneously uses all (J) cumulative logits. The simple form of the model is denoted by:

$$logit[P(Y \leq j|\boldsymbol{x})] = \alpha_j + \boldsymbol{\beta}^T\boldsymbol{x}, \quad j = 1, 2, \ldots, J-1 \, ;$$

Where:

$\alpha_j$ = Separate intercept parameters also known as cut-off parameters
$j$ = Level of the ordered category with 3 levels
$\beta^T$ = Sets of regression parameters for each logit
$\boldsymbol{x}$ = The set of explanatory variables (1).

Every cumulative logit has an intercept that increases as the categories of the dependent variable increase. The model assumes the same effects of β for each logit [13, 14]. The odds ratio with their 95% confidence interval was calculated to find out the magnitude and relationship presence. The proportionality of odds for the dependent variable was tested using the Chi-square test and log-likelihood test.

## Bongaarts model of proximate determining factors of fertility

The five factors that symbolize teenage fertility in the Bongaarts framework include, portion married ($C_m$), Contraception ($C_c$), induced abortion ($C_a$), postpartum infecundability ($C_i$), and primary sterility ($C_p$). If an index is close to 1, the fertility delay effect is insignificant. However, if an index is 0, its effect on fertility impediment is maximum. The formula below gives a summary of the Bongaarts original model [11, 15]. It is represented as: -

$$\textbf{TFR} = \textbf{C}_m * \textbf{C}_c * \textbf{C}_a * \textbf{C}_i * \textbf{C}_p * \textbf{TF} \tag{1}$$

TFR represents the Total Fertility Rate; the average number of children a woman will have born by the end of her reproductive life (approximately 50 years). While TF represents the Total Fecundity Rate; the total number of births to a woman by the end of her reproductive life if she was to remain married and according to the prevalent age-specific marital fertility rate. The aggregate model above can be modified for use in establishing age-specific fertility determinants. In addition, the model is applied even if some index information is missing or unavailable by assuming the value 1 for those indices. Consequently, since abortion is illegal in Kenya, the abortion index was equated to 1 [11, 15]. Replacing the mathematical Eq (1) above, the analysis of age-specific fertility determinants for the 15–19 years adolescents becomes: -

$$\textbf{ASFR} = \textbf{C}_{m\ (15-19)} * \textbf{C}_{c\ (15-19)} * \textbf{C}_{a\ (15-19)} * \textbf{C}_{i\ (15-19)} * \textbf{C}_{p\ (15-19)} * \textbf{AF} \tag{2}$$

$C_{m\ (15-19)}$, $C_{c\ (15-19)}$, $C_{a\ (15-19)}$, $C_{i\ (15-19)}$, and $C_{p\ (15-19)}$ represents indices between 15–19 years computing fertility impeding effect of marriage, contraception, abortion, postpartum insusceptibility and sterility index correspondingly. Further AF, represents the age-specific fecundity rate, the highest potential biological number of births, while ASFR is the Age-Specific Fertility Rate. Since AF is equal to 511 per 1000 women aged between 15 and 19 years, 2.5 births per woman are arrived at by multiplying AF by five [11]. Theoretically, it means, a teenager who did not breastfeed, stayed married from 15 to 19 years, in addition to not using contraceptives the highest possible births would be 2.5 by age 19 years. Replacing AF with 2.5 in Eq (2) above becomes:

$$\textbf{ASFR} = \textbf{C}_{m\ (15-19)} * \textbf{C}_{c\ (15-19)} * \textbf{C}_{a\ (15-19)} * \textbf{C}_{i\ (15-19)} * \textbf{C}_{p\ (15-19)} * \textbf{2.5} \tag{3}$$

Eq 3 will thus be used to calculate the indices for the proximate variables stated above.

**Calculation of marriage ($C_{m (15-19)}$) index.** Marriage index is calculated with the formula: $C_{m\ (15-19)} = m\ (20-24)*0.75$. In the equation, m (20–24) is the fraction of married women among females aged 20–24 years. Since premarital conception occurrences among teenagers between 15 to 19 years are significant, 0.75 is a constant in the equation.

**Calculation of contraception ($C_c (15-19)$) index.** Contraception index is expressed by; Cc (15–19) = 1-c*u/f. In the formula, c is typically 0.61 for women aged 24 years and below, representing the mean contraception effectiveness while u denotes the presently married fraction of adolescents who were also presently using contraception. f is typically 0.98 for women aged 24 years and below, symbolizing the fraction of presently fecund women.

**Calculation of postpartum insusceptibility ($C_i (15-19)$) index.** Postpartum insusceptibility index will be calculated by the formula:

$$\textbf{C}_{i\ (15-19)} = \frac{20}{18.5 + i\ (15-19)}$$

In the formula, i (15–19) denotes months of postpartum insusceptibility which is derived from the mean length of breastfeeding. The value of 'i' is equal to $1.753e^{0.1396B-0.00187B*B}$. In the equation, B denotes the average breastfeeding period in months.

**Primary sterility (Cp (15–19)) index estimation.** The primary sterility index was assigned the value of 1 since its effect is insignificant in countries of Eastern Africa, Kenya included [6].

**Index of induced abortion (Ca (15–19)) estimation.** In Kenya, abortion is illegal. Consequently, the value of the index will be 1; meaning abortion has a negligible effect on fertility [15].

### Ethics statement

Specific ethical approval was not required for the 2014 KDHS secondary data analysis. Consequently, a secondary analysis was done under the original consent provided by participants during the data collection process. However, permission to use the data was obtained from ICF Macro from the URL https://dhsprogram.com/data. The dataset title is KEIR7SV.ZIP. Their user instructions were followed, which included, treating the data as confidential and no effort should be made to identify any household or individual respondent interviewed in the survey.

## Results

### Sample characteristics

Table 1 presents basic demographic, economic, and cultural characteristics of adolescents who were 5,820. A majority, 3,510 (60.3%) were young adolescents between 15–17 years of age and the rest 2,309 (39.7%) were 18 to 19 years of age. A majority of the adolescents, 1,230 (57.6%) had their first sex between 15–17 years of age, and most of them, 5,210 (89.5%) had not been married. Most adolescents (68.1%) lived in rural areas and they were neither working 2,108 (77.6%) nor using contraceptives 5,232 (89.9%). Those who had attained primary education were the majority (49.9%) followed by secondary education (45%) while a few had attained higher education (2.8%) and the rest had no education (2.3%). Approximately 36.7% of the adolescents who were interviewed had engaged in sexual activities with a majority (86.7%) having their first intercourse aged 17 years and below and as a result, they had given birth more compared to those whose first intercourse age was from 18 to 19. Additionally, 28.2% of adolescents aged between 18 and 19 years had given birth compared to only 5.8% of adolescents aged between 15–17 years. Further, adolescents who were married, working, and using contraceptives had more births than their counterparts. About, 29.4% of adolescents with no education had given birth compared to 19.6% who had primary education and 9.1% who had attained secondary education. In addition, 26.8% of adolescents who were of the Maasai/Samburu ethnic group had the most births, followed by Luo (19.8%), Mijikenda/Swahili/Taita/Taveta (17.9%), and Kalenjin/Turkana (17.5%) respectively. Significant explanatory factors that were related to adolescent fertility at a 5% significance level included age at first intercourse, current age, current marital status, highest education level attained, wealth index, employment status, current contraceptive use, religion, regularity of listening to the radio and ethnicity.

Table 2 describes the adolescent's fertility patterns. The average age was 16.5 years. Virtually, 855 (14.7%) had given birth while 4965 (85.3%) had not. Among the adolescents who had ever given birth, approximately three quarters (74.7%) had first birth at 17 years or below whereas 560 (65.5%), had first birth amid 15 to 17 years. Twelve (12) years was the youngest age at birth stated. The mean year of the adolescent at first birth was 16.5 ± 1.5 years while the

**Table 1. Variables distribution and association of adolescents fertility in Kenya, 2014.**

| Variable (n = 5820) | Frequency | % | Children Ever Born (%) | | | |
| --- | --- | --- | --- | --- | --- | --- |
| | | | 0 | 1 | 2+ | P-Value |
| Age at first sex | | | | | | < 0.0001* |
| 18–19 | 283 | 13.3 | 82.7 | 17.3 | 0.0 | |
| 15–17 | 1230 | 57.6 | 59.2 | 35.9 | 5.0 | |
| <15 | 622 | 29.1 | 55.1 | 32.0 | 12.9 | |
| Current age | | | | | | 0.000* |
| 18–19 | 2309 | 39.7 | 71.8 | 22.4 | 5.8 | |
| 15–17 | 3510 | 60.3 | 94.2 | 5.4 | 0.4 | |
| Current Marital Status | | | | | | < 0.0001* |
| Not Married | 5210 | 89.5 | 91.9 | 7.4 | 0.7 | |
| Married | 609 | 10.5 | 29.2 | 52.5 | 18.2 | |
| Highest education level | | | | | | < 0.0001* |
| Higher | 165 | 2.8 | 93.9 | 6.1 | 0.0 | |
| Primary | 2903 | 49.9 | 80.4 | 15.4 | 4.2 | |
| Secondary | 2620 | 45.0 | 91.0 | 8.5 | 0.6 | |
| No education | 133 | 2.3 | 70.7 | 21.1 | 8.3 | |
| Type of place of residence | | | | | | 0.179 |
| Rural | 3961 | 68.1 | 85.0 | 12.2 | 2.8 | |
| Urban | 1859 | 31.9 | 86.0 | 12.0 | 2.0 | |
| Wealth Index | | | | | | < 0.0001* |
| Rich | 2228 | 38.3 | 89.4 | 9.3 | 1.3 | |
| Middle | 1332 | 22.9 | 84.2 | 13.7 | 2.1 | |
| Poor | 2260 | 38.8 | 81.9 | 14.0 | 4.1 | |
| Employment Status | | | | | | < 0.0001* |
| Working | 610 | 22.4 | 71.5 | 22.5 | 6.1 | |
| Not Working | 2108 | 77.6 | 89.6 | 8.8 | 1.6 | |
| Current Contraceptive Use | | | | | | < 0.0001* |
| Using | 588 | 10.1 | 34.9 | 53.2 | 11.9 | |
| Not using | 5232 | 89.9 | 91 | 7.5 | 1.5 | |
| Religion | | | | | | < 0.0001* |
| Other/No religion | 52 | 0.9 | 50.0 | 36.5 | 13.5 | |
| Protestant | 4062 | 69.9 | 85.7 | 11.7 | 2.6 | |
| Muslim | 485 | 8.3 | 87.0 | 9.3 | 3.7 | |
| Roman Catholic | 1213 | 20.9 | 84.9 | 13.8 | 1.3 | |
| Frequency of Listening to Radio | | | | | | 0.002* |
| Less than once a week | 922 | 15.9 | 86.6 | 12.1 | 1.3 | |
| At least once a week | 3842 | 66.0 | 85.9 | 11.3 | 2.7 | |
| Not at all | 1053 | 18.0 | 82.1 | 15.0 | 2.8 | |
| Ethnicity | | | | | | < 0.0001* |
| Other | 146 | 2.5 | 83.6 | 12.3 | 4.1 | |
| Kalenjin /Turkana | 816 | 14.0 | 82.5 | 14.3 | 3.2 | |
| Kamba | 653 | 11.2 | 89.9 | 9.3 | 0.8 | |
| Embu /mbeere/Meru | 331 | 5.7 | 84.3 | 14.5 | 1.2 | |
| Kisii | 321 | 5.5 | 85.0 | 13.1 | 1.9 | |
| Luhya/Iteso | 1089 | 18.7 | 84.7 | 12.9 | 2.5 | |
| Luo | 726 | 12.5 | 80.2 | 16.5 | 3.3 | |
| Maasai/Samburu | 149 | 2.6 | 73.2 | 17.4 | 9.4 | |

(*Continued*)

**Table 1.** (Continued)

| Variable (n = 5820) | | | Children Ever Born (%) | | | |
|---|---|---|---|---|---|---|
| | Frequency | % | 0 | 1 | 2+ | P-Value |
| Mijikenda/Swahili/Taita/Taveta | 436 | 7.5 | 82.1 | 13.5 | 4.4 | |
| Boran/Gabbra/Somali | 206 | 3.5 | 91.7 | 4.9 | 3.4 | |
| Kikuyu | 946 | 16.3 | 92.1 | 6.7 | 1.3 | |

Source: *own calculations;* *Significant at 5%; P≤0.05.*

median age was 17 years. Overall, 2.6% of the adolescents had given birth to two or more children while 12.1% had given birth to 1 child. Average births were 1.2 ± 0.4. Nearly 263 (4.5%) of teenagers were pregnant during the survey period.

## Proximate determinants of adolescent fertility

Table 3 illustrates proximate factors associated with adolescent childbearing in Kenya. The observed age-specific fertility rate (ASFR) was 0.17 births per woman. Implying, nearly 2.33 births per adolescent were prevented because of not being married, use of contraception, and postpartum infecundability. Close to 2.34 births per adolescent out of the highest biological number of 2.5 births were avoided among urban teenagers whereas 2.32 births per adolescent residing in rural areas were prevented. Adolescents' marital index (Cm) was 0.36, implying, not being married decreased adolescent childbearing by 64 percent. The marital indicator was lesser for adolescents that resided in urban areas (0.34) compared to adolescents that resided in rural areas (0.37). Postpartum infecundability was also a significant factor associated with adolescent childbearing; overall it decreased 24% of the biologically maximum expected adolescent childbearing level in marriage. Its impact was rather greater among teenage urban

**Table 2.** Adolescent reproductive pattern in Kenya, 2014.

| Characteristic | Percentage (%) |
|---|---|
| **Ever given Birth(n = 5820)** | |
| Yes | 14.7 |
| No | 85.3 |
| **Pregnant Currently (n = 5820)** | |
| Yes | 4.5 |
| No | 95.5 |
| **Total Children Ever Born (n = 5820)** | |
| 2+ | 2.6 |
| 1 | 12.1 |
| 0 | 85.3 |
| Average birth ± SD = 1.2 ± 0.438 | |
| **Age at first birth in Years (n = 855)** | |
| <15 | 9.2 |
| 15–17 | 65.5 |
| 18–19 | 25.3 |
| Average age ± SD = 16.5±1.48 | |
| Intermediate Age in years = 17 | |

Source: *own calculations.*

**Table 3. Proximate determinants indices of adolescent fertility, Kenya, 2014.**

| Index (n = 5820) | Rural (index) | Births Averted | Urban (index) | Birth Averted | Total (Index) | Births Averted (Total) |
|---|---|---|---|---|---|---|
| Marital index ($C_m$) | 0.37 | 2.13 | 0.34 | 2.16 | 0.36 | 2.14 |
| Contraception Use index ($C_c$) | 0.78 | 1.72 | 0.68 | 1.82 | 0.75 | 1.75 |
| Postpartum Insusceptibility index ($C_i$) | 0.77 | 1.73 | 0.74 | 1.76 | 0.76 | 1.74 |
| Predicted ASFR* | 0.56 | 1.94 | 0.43 | 2.07 | 0.51 | 1.99 |
| Observed ASFR** | 0.18 | | 0.16 | | 0.17 | |

*Predicted by Bongaarts Index.

** Calculated from births in the last five years.

dwellers compared to adolescents who were rural dwellers. The use of contraception decreased fertility among adolescents by 25% of the total marital fertility. The effect was stronger in urban Kenya with a reduction of 32% of all marital fertility compared to rural areas where the reduction was 22%. Family planning use is more effective in urban Kenya compared to rural Kenya.

## Factors associated with adolescents fertility

Table 4 shows the association of selected independent variables with the outcome variable of adolescent fertility. The ordinal regression model was fitted based on predictor variables that were found to be significant with the outcome variable after bivariate analysis and the most important variables retained for the regressions analysis. Overall, age at first intercourse, current age, the current status of marriage and current contraceptive usage were important determinants of teenage childbearing. Adolescents who had first intercourse from age 18 and above had fewer children (OR: 0.221; 95% CI: 0.124–0.392) and so do adolescents that had first sex between age 15 and 17 (OR: 0.530; 95% CI: 0.379–0.742) in comparison with those that had first sex when they were less than 15 years old. Older adolescents (18–19 years) were associated with higher fertility. Adolescents who were between 18 and 19 years of age had more children (OR: 4.727; 95% CI: 3.318–6.733) than those who were between 15 and 17 years old. Adolescent non-marriage was associated with lower fertility. Unmarried adolescents had a lower number of children (OR: 0.212; 95% CI: 0.150–4.780) than those who were married. In addition, contraceptive use among adolescents was associated with increased fertility with an adolescent who was using contraceptive having a higher number of children (OR: 3.138; 95% CI: 2.257–4.362) than those who were not using a contraceptive. On the other hand, the highest education level attained, wealth index, employment status, religion, regularity of radio listening and ethnicity were insignificant after controlling for other factors.

## Discussion

Generally, most of the adolescents interviewed had first sex between 15–17 years of age, were not married, had attained primary education (49.9%) others had managed secondary education (45%) as well as not using contraceptives (89.9%). In addition, most of them had not given birth (85.3%). For those who had given birth (14.7%), a majority's age at first birth was between 15–17 years. This was consistent with the finding of the study by [2] which revealed that almost one in five female adolescents had a child before their 18th birthday and the fact that a majority of them (9 out of 10) were not married.

It was observed that 85.3% of female adolescents interviewed had not given birth while 14.7% have given birth. This was consistent with an analysis by [2] that revealed that globally,

**Table 4. Determinants of adolescents fertility in Kenya, 2014.**

| | | 95% CI | | |
|---|---|---|---|---|
| Variable (n = 5820) | AOR | Lower | Upper | p-value |
| Age at First Sex | | | | |
| 18–19 | 0.221 | 0.124 | 0.392 | < 0.000* |
| 15–17 | 0.53 | 0.379 | 0.742 | < 0.000* |
| <15 (RC) | | | | |
| Current Age | | | | |
| 18–19 | 4.727 | 3.318 | 6.733 | < 0.000* |
| 15-17(RC) | | | | |
| Current Marital Status | | | | |
| Not Married | 0.212 | 0.15 | 4.78 | < 0.000* |
| Married(RC) | | | | |
| Highest Education level | | | | |
| Higher | 0.374 | 0.093 | 1.498 | 0.165 |
| Primary | 1.555 | 0.529 | 4.566 | 0.422 |
| Secondary | 0.755 | 0.248 | 2.299 | 0.621 |
| No education(RC) | | | | |
| Wealth Index | | | | |
| Rich | 0.69 | 0.469 | 1.015 | 0.06 |
| Middle | 0.877 | 0.594 | 1.296 | 0.511 |
| Poor(RC) | | | | |
| Employment Status | | | | |
| Working | 1.091 | 0.801 | 1.485 | 0.580 |
| Not Working (RC) | | | | |
| Current Contraceptive Use | | | | |
| Using | 3.138 | 2.257 | 4.362 | < 0.000* |
| Not Using(RC) | | | | |
| Religion | | | | |
| Other/No religion | 5.608 | 1.55 | 20.289 | 0.009* |
| Protestant | 1.047 | 0.726 | 1.509 | 0.808 |
| Muslim | 1.298 | 0.463 | 3.643 | 0.620 |
| Roman Catholic(RC) | | | | |
| Regularity of Radio Listening | | | | |
| < once a week | 0.704 | 0.408 | 1.215 | 0.207 |
| At least once a week | 1.032 | 0.685 | 1.554 | 0.880 |
| Not at all(RC) | | | | |
| Ethnicity | | | | |
| Other | 0.887 | 0.277 | 2.84 | 0.840 |
| Kalenjin /Turkana | 1.51 | 0.851 | 2.68 | 0.159 |
| Kamba | 0.756 | 0.388 | 1.475 | 0.413 |
| Embu /mbeere/Meru | 0.9 | 0.419 | 1.933 | 0.787 |
| Kisii | 1.643 | 0.79 | 3.417 | 0.184 |
| Luhya/Iteso | 1.716 | 0.972 | 3.03 | 0.063 |
| Luo | 1.335 | 0.752 | 2.369 | 0.324 |
| Maasai/Samburu | 2.24 | 0.885 | 5.67 | 0.089 |
| Mijikenda/Swahili/Taita/Taveta | 1.664 | 0.709 | 3.904 | 0.242 |
| Boran/Gabbra/Somali | 0.742 | 0.148 | 3.731 | 0.717 |

Source: own calculations. RC = Reference Category; *Significant at p<0.05.

teenagers are not necessarily having more children since adolescent fertility has slightly declined. Nevertheless, among the adolescent births that do occur, more occur outside marriage. Subsequently, the biggest challenge in Kenya is the mistimed births among fertile adolescents. The overall country-specific proportion of adolescents that had given birth in Kenya was lower than the proportion documented for Malawi (20.1%), Uganda (19.2%), Tanzania (19.6%) and higher than the proportion documented in Ethiopia (14.4%), Rwanda, (3.3%), Eritrea (11.0%) and Ghana (10.2%) [8, 16–18].

An analysis of the Bongaarts model revealed that fertility proximate variables had a greater influence on biological fertility decline among adolescent urban dwellers than rural dwellers. In addition, not being married resulted in a 64% reduction in observed ASFR ($C_m$ = 0.36). The probable explanation is that a majority (89.5%) were not married. Contraceptive usage had an inhibiting result of 25%. Comparatively, fertility obstruction for usage of contraceptive, not being married and postpartum infecundity was greater among urban dwellers ($C_c$ = 0.68, $C_m$ = 0.34, $C_i$ = 0.74) than rural dwellers (Cc = 0.78, Cm = 0.37, Ci = 0.77). The variation between the two categories is caused by the fact that the probability of adolescents who reside in urban areas, using contraceptives and delaying marriage is higher compared to their rural counterparts [6]. However, the index of postpartum infecundity contradicts the results of an Ethiopian study since these study findings revealed that the inhibitory effect of postpartum infecundity was greater among adolescent urban residents compared to rural dwellers [19]. Generally, the predicted ASFR for both urban and rural residences was substantively greater than the observed age-specific fertility rate (0.51 versus 0.17). The greater variation observed while comparing model and observed estimates would be attributed to exclusion of significant determinants in the regression [6]. This may include underreporting of contraceptive use and absence of abortion from the model, resulting in an overestimation of adolescent fertility [19, 20].

In the study, age at first sex, current age, marital status and use of contraceptives were the key factors associated with adolescent childbirth in Kenya. Teenagers who had had their first sex in late adolescence (18–19 years) had a lesser probability of giving birth. The finding is in line with that of Ethiopia by [21] which revealed that fertility was higher for adolescents who began sexual intercourse before their 18[th] birthday compared to those who had not begun sexual intercourse.

Adolescent age was a significant determinant of fertility. Older adolescents (18–19) had a higher probability of fertility compared to the younger ones. This was in line with by most of the research inferences that indicate a positive relationship between adolescent age and fertility. Some of the studies include a Malawian study utilizing 2010 Malawi DHS data that revealed an increase in teenage fertility with rising adolescent age. This was also consistent with findings of Brazil, Nigeria, and Ethiopia [6, 9, 22, 23].

The adolescent state of marriage was an important determinant of adolescent childbearing. Adolescents who were not married had a lesser probability of having children unlike the married. Results of the study revealed that the percentage of married adolescents who had children was 70.7%, while the percentage of adolescents who were not married and had children was 8.1%. Implying, married adolescents had a higher likelihood of giving birth unlike unmarried adolescents. It is worth noting that despite marriage being the most important proximate determinant of fertility, being pregnant can also accelerate propensity to marry, hence a limitation of this study that needs to be examined. The finding was consistent with studies in Malawi, Ethiopia, Nigeria and Central Java which revealed that marital status had a positive effect on fertility [7, 9, 10, 23].

Contraceptive use was a significant determinant of adolescent fertility. Adolescents who were using contraceptives had a greater chance of having given birth. This was in line with

other African studies which found out that teenagers who were currently using contraceptives had higher fertility [7, 23].

## Strengths

The study utilized data from the Kenya Demographic and Health Survey, 2014. The sample for the survey was drawn from a master sampling frame, the Fifth National Sample Survey and Evaluation Programme (NASSEP V) which was weighted for national representation. Consequently, the sample was a representative estimate of the entire adolescent population in the country. In addition, sampling ensured more detailed information on factors associated with adolescent fertility was collected with more accuracy and reliability. Thus, the data being nationally representative, accurate and reliable, the findings of the study gives insights that will enhance the design and implementation of reproductive health strategies, policies and programs aimed at reducing adolescent childbearing in Kenya.

## Study limitation

The study utilized secondary data from 2014 KDHS; hence, the study relied on variables collected during the survey. Further, since the DHS data is cross-sectional we could only ascertain associations and not causality for the predictor variables under study. In addition, since abortion is illegal in Kenya, data on induced abortion was not collected, hence the index was assigned the value 1. Similarly, the index on primary sterility was considered as 1 as its effect in Kenya is insignificant. Given that no similar survey has been conducted since 2014, the data could not provide analytical insights into the prevailing adolescent childbearing rate, given anecdotal evidence that teenage childbearing increased during the closure of schools due to the COVID-19 pandemic in 2020.

## Conclusion

The results of the analysis of the Bongaarts model revealed that not being married was the single most important factor associated with adolescent childbearing in Kenya. On fitting the ordinal regression model, the results concluded that age at first intercourse, current age, marital status, and usage of contraceptives were the key contributing factors to adolescent childbearing in Kenya. Teenagers who had had initial intercourse in late adolescence the (18–19 years) had a lower likelihood of giving birth. In addition, older adolescents (18–19) had a greater probability of fertility, unlike younger adolescents. Further, adolescents who were using contraceptives were more likely to have given birth unlike non-users. Whereas unmarried teenagers had a lower probability of fertility compared to those who were married.

High adolescent fertility in Kenya remains a great challenge. The government should strive to increase secondary school enrollment and discourage early marriages before the age of 18 in the country. In addition, targeted programs should be developed to delay age at first sex which has an inverse relationship to adolescent childbearing in the country. The programs should be launched equally among adolescents dwelling in rural and urban regions since adolescent's residence was not an important factor affecting fertility.

## Acknowledgments

We are grateful to Demographic and Health Surveys for providing access to the data.

## Author Contributions

**Conceptualization:** Naomi Monari, Alfred Agwanda.

**Data curation:** Naomi Monari.

**Formal analysis:** Naomi Monari.

**Funding acquisition:** Naomi Monari.

**Methodology:** Naomi Monari, Alfred Agwanda.

**Project administration:** Naomi Monari.

**Resources:** Naomi Monari.

**Software:** Naomi Monari.

**Supervision:** Alfred Agwanda.

**Writing – original draft:** Naomi Monari.

**Writing – review & editing:** James Orwa, Alfred Agwanda.

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
