## [Decision Letter · Decision Letter 0]

13 Jul 2021

PONE-D-21-14209

Adolescent Fertility and its Determinants in Kenya: Evidence from Kenya Demographic and Health Survey 2014.

PLOS ONE

Dear Dr. NAOMI MORAA MONARI,

Thank you for submitting your manuscript to PLOS ONE. After careful consideration, we feel that it has merit but does not fully meet PLOS ONE’s publication criteria as it currently stands. Therefore, we invite you to submit a revised version of the manuscript that addresses the points raised during the review process.

We look forward to receiving your revised manuscript.

Kind regards,

Shah Md Atiqul Haq

Academic Editor

PLOS ONE

Journal Requirements:

Additional Editor Comments:

Dear Authors,

I would ask you to revise the paper by following the reviewers' comments and suggestions.

Reviewers' comments:

Reviewer's Responses to Questions

**Comments to the Author**

1. Is the manuscript technically sound, and do the data support the conclusions?

Reviewer #1: Partly

Reviewer #2: Yes

2. Has the statistical analysis been performed appropriately and rigorously? 

Reviewer #1: Yes

Reviewer #2: Yes

3. Have the authors made all data underlying the findings in their manuscript fully available?

Reviewer #1: Yes

Reviewer #2: Yes

4. Is the manuscript presented in an intelligible fashion and written in standard English?

Reviewer #1: Yes

Reviewer #2: Yes

5. Review Comments to the Author

Reviewer #1: The text is understandable, but before publishing it anywhere, it should be edited for English.

This is an elementary analysis with very little new information. I think the importance of marriage is probably misleading, since I suspect that a great number of these were premarital pregnancies. Also the arguments for the C Index values are not convincing.

Reviewer #2: Thank you for giving me the opportunity to review this interesting article. As it has been noted, there is a dearth of recent literature exists addressing determinates of adolescents’ fertility in the context of Kenya. I hope my comments would be helpful to increase the quality of your work.

Abstract

• Please be specific on age 15-17 years old, it could be clarified as “current age 15-17 years “

• Add few more Key words which are relevant to your present study.

Background

• Do they have any recent studies on the determinates of fertility in the context of Kenya?

• It is not clear what do you really mean “These factors were varied from country to country and even within the 22 countries disparities still exist”, what sort of inequalities? Need a justification.

• There should be a proper discussion on what are the current research gaps in terms of fertility determinants in Kenya, it seems like jumping to the objectives without proving enough evidence/research gaps, what is the rationale and how this study could be benefited from bridge the gaps, and what would be the contribution of present study- one brief paragraph looking at this should be included.

• This sentese is not clear “Exploring the factors associated with teenage childbearing ensures targeted programs to this vulnerable population sub-group for policy makers and to aid in reducing the teen pregnancy incidence” So have you used the terms “teenage and adolescents” interchangeably?

Methods

• Methods section is clear; however, you may add an abbreviation terms somewhere for the convenience of the reader.

• Dependent variables need to be clearer, particularly, how you have been developed some of the covariates such as wealth index, etc, do they compatible with DHS?

Results:

• For all Tables include the sample size (n=)

• As according to the Table 3, as it has presented the disaggregation of fertility indexes by urban rural, what is the intention of doing so, this seems arbitrary, as this has not mentioned in the objectives.

Discussion

• So, do you mean that even after 20 years of Bledsoe and Cohen, 1993 analysis, still the fertility figures are same? Justify this.

• Discussion and conclusion sections are clear, it would be better to come up with separate section on strengths and limitations.

Acknowledgement

Why the author mentioned as “we”? Do you have any other author contributions?

6. PLOS authors have the option to publish the peer review history of their article (what does this mean?). If published, this will include your full peer review and any attached files.

Reviewer #1: No

Reviewer #2: **Yes: **Gayathri Abeywickrama

---

## [Author Response · Author response to Decision Letter 0]

27 Oct 2021

We wish to thank the academic editor and the two reviewers for their useful comments on our manuscript. We hope that we have satisfactory tackled all concerns raised and that the manuscript is now well suited for publication.

---

## [Editor Report · Decision Letter 1]

16 Dec 2021

Adolescent Fertility and its Determinants in Kenya: Evidence from Kenya Demographic and Health Survey 2014.

PONE-D-21-14209R1

Dear Dr.NAOMI MORAA MONARI,

We’re pleased to inform you that your manuscript has been judged scientifically suitable for publication and will be formally accepted for publication once it meets all outstanding technical requirements.

Kind regards,

Shah Md Atiqul Haq

Academic Editor

PLOS ONE

Additional Editor Comments (optional):

Dear authors,

Your paper is now accepted.
---

## [Editor Report · Acceptance letter]

23 Dec 2021

PONE-D-21-14209R1 

Adolescent Fertility and its Determinants in Kenya: Evidence from Kenya Demographic and Health Survey 2014. 

Dear Dr. Monari:

I'm pleased to inform you that your manuscript has been deemed suitable for publication in PLOS ONE. Congratulations! Your manuscript is now with our production department. 

Kind regards, 

on behalf of

Dr. Shah Md Atiqul Haq 

Section Editor

PLOS ONE